# Are There Non-Invasive Biomarker(s) That Would Facilitate the Detection of Ovarian Torsion? A Systematic Review and Meta-Analysis

**DOI:** 10.3390/ijms252111664

**Published:** 2024-10-30

**Authors:** Meg Naylor, Grace Doherty, Hannah Draper, Daniel M. Fletcher, Alan Rigby, Tolu Adedipe, Barbara-ann Guinn

**Affiliations:** 1Hull York Medical School, University of York, York YO10 5DD, UK; hymn17@hyms.ac.uk; 2Centre for Biomedicine, Hull York Medical School, University of Hull, Hull HU6 7RX, UK; g.e.doherty-2020@hull.ac.uk (G.D.); d.m.fletcher-2018@hull.ac.uk (D.M.F.); 3Hull York Medical School, University of Hull, Hull HU6 7RX, UK; alan.rigby@hyms.ac.uk; 4Women and Children’s Hospital, Hull University Teaching Hospitals NHS Trust, Anlaby Road, Hull HU3 2JZ, UK; busolade@gmail.com

**Keywords:** ovarian torsion, biomarker, non-invasive, blood, IL-6, systematic review

## Abstract

Ovarian torsion (OT) is a rare gynaecological emergency that requires a prompt diagnosis for optimal patient management. To determine whether there were any biomarkers suitable for the non-invasive detection of OT, two independent reviewers performed systematic searches of five literature databases (PubMed, Medline, Scopus, Cochrane, and CINAHL) from inception until October 1st, 2023. Following the Preferred Reporting Items for Systematic reviews and Meta-Analyses (PRISMA) guidelines, the search included patients with OT that had quantified biomarker expression with no age, geographical location, publication date, language, or setting restrictions. Articles were excluded if OT was found incidentally, was based on qualitative analyses, or were not primary research articles. Full texts of 23 selected articles were assessed for risk of bias and quality assurance using a modified Newcastle–Ottawa Scale (NOS) for clinical studies and SYRCLE’s risk of bias tool for the assessment of pre-clinical (animal) studies. A total of 11 articles described studies on animals and all described serum biomarkers comparing results between OT versus a sham operation, a control group, or readings before and after OT. Ischaemia-modified albumhumin (IMA), serum D-dimer (s-DD), heat shock protein-70 (hsp-70), Pentraxin-3 (PTX3), and c-reactive protein (CRP) each showed the most promise, with *p*-values for the difference between OT and control groups achieving ≤ 0.001. In studies of humans, the biomarkers ranged from 16.4 to 92.3% sensitivity and 77–100% specificity. The most promising biomarkers for the early prediction of OT in patients included s-DD, interleukin-6 (IL-6), IMA, and tumour necrosis factor-alpha (TNF-α). Signal peptide, CUB domain, and EGF-like domain-containing 1 (SCUBE1) had a high specificity at 93.3%, second only to s-DD and a positive likelihood ratio (LR) > 10. IMA was the only other biomarker that also had a positive LR > 10, making it a promising diagnostic biomarker. The studies identified by this systematic literature review each analysed small patient groups but IMA, DD, and SCUBE1 nevertheless showed promise as serum biomarkers with a pooled LR > 10. However, further well-designed studies are needed to identify and evaluate individual markers or diagnostic panels to help clinicians manage this important organ-threatening condition.

## 1. Introduction

Ovarian torsion (OT) is a rare gynaecological emergency that requires prompt surgical intervention to prevent ovarian ischaemia, necrosis, and loss of function. OT affects approximately 9.9/100,000 women of reproductive age each year [1]. OT arises when the ovary twists over its supporting ligaments in the adnexa [2], and in combination with the fallopian tube, is termed an adnexal torsion (AT) (Figure 1). OT is most commonly associated with benign cysts greater than 5 cm [3], but 20% occur in pre-pubescent girls, 50% of whom have normal-sized ovaries but may have elongated infundibulopelvic ligaments [4]. People who are pregnant or undergoing fertility treatments are at particular risk of OT due to enlarged follicles on the ovary [5].

OT frequently presents with symptoms such as acute onset pain, nausea, and vomiting [6], making differentiating OT from acute abdomen difficult. Ultrasound (US) is reported to have a sensitivity of 84% and common findings include large oedematous ovaries, free pelvic fluid, and the “whirlpool” sign (WS) arising from twisted vascular pedicles [4]. Surgical intervention, normally in the form of diagnostic laparoscopy, is the gold standard for the diagnosis of OT; furthermore, it enables concurrent treatment. In premenopausal women, this is now routinely undertaken based on clinical findings even when the US is normal to preserve ovarian function [7]. Novoa et al. [8] reported that only one in five necrotic-appearing ovaries were confirmed histopathologically [7]. An oophorectomy may still need to be performed when there is a significant diagnostic delay or in the presence of other complicating factors. The consequences of this remain uncertain [9], although the impact on quantity but not quality of the ovarian reserve can adversely affect women seeking assisted reproductive techniques [10].

The risk of a serious complication from a gynaecological laparoscopy [11] is approximately 2 per 1000 but can include organ damage or major vessel injury, both of which are associated with significant morbidity and mortality [12]. The long-term consequences of bilateral oophorectomy are increasingly understood and may include the risk of all-cause mortality, coronary artery diseases, and non-gynaecological cancers [13,14,15,16,17]. The need to preserve the ovaries should therefore be considered for reasons more than just maintaining reproductive function.

Biomarkers can play a crucial role in both enabling early diagnosis and avoiding unnecessary investigations. They have the advantage of being a non-invasive intervention and their use is already well established for a number of key medical and surgical pathologies, including bowel cancer (CEA) [18], pancreatitis (serum amylase) [19], and prostate cancer (PSA) [20]. Yet, despite the severity of OT and likely due to the non-specific symptoms that can mimic other conditions, the lack of disease-specific blood biomarkers, and its relative scarcity, there are thus far no non-invasive biomarkers that can reliably aid its diagnosis. Potential diagnostic biomarkers for OT have been identified, although few have evidence to support their use in clinical practise. This study aims to compare non-invasive biomarkers for the detection of OT identified in animal models and human studies and to determine their potential to provide an early non-invasive indication of OT.

## 2. Materials and Methods

### 2.1. Systematic Review

The Preferred Reporting Items for Systematic reviews and Meta-Analyses (PRISMA) guidelines were adhered to [21,22], including the development of a protocol (Appendix A) and prospective registration. In accordance with the Population Intervention Comparator Outcome (PICO) framework, we formulated our research question, which was “Are there non-invasive biomarker(s) that would facilitate the detection of ovarian torsion?”. In the animal studies, the population was rats with surgically induced OT, the intervention involved non-invasive biomarkers that could be used to predict OT, comparators were rats without torsion, and the outcome was whether the biomarker level was raised. In human studies, the population was adults with OT, interventions were non-invasive biomarkers that predicted OT, comparators were patients with OT symptoms which were not confirmed, and the outcome was timely treatment to save the ovary and indications of no negative impact on fertility.

The search strategy was developed based on index terms that were found in three to six sentinel articles that were identified following an initial screen of the literature using PubMed. We identified articles in five online databases (PubMed, Medline, Scopus, Cochrane, and CINAHL) from inception until October 1st, 2023. Published manuscripts focusing on OT and non-invasive biomarkers were identified using MeSH search terms as follows: biomarker* OR “biological marker*” OR “metabolic process” OR “disease diagnosis” OR “molecular marker*” OR “signature molecule*” OR “bio* indicator*” OR “blood indicator*” Blood OR “blood sample*” OR “blood analysis” AND “Ovar* torsion” OR “Adnexal torsion”.

For the purpose of the literature search, two independent reviewers (G.D. and M.N.) used prespecified inclusion/exclusion criteria (Table 1) to screen articles based on the title and abstract. At the stage of abstract reading, the search excluded books, systematic reviews, meta-analyses, and conference papers. Studies assessed were not limited by language and included patients with OT that had quantified biomarker expression, with no age, geographical location, publication date, or setting restrictions (Table 1). Articles were excluded if OT was found incidentally or were based on qualitative analyses.

Duplicate studies were removed and the articles screened by title and abstract. The full text of the studies remaining were assessed against the inclusion criteria to determine their appropriateness for this review.

Review articles were only removed once the cited articles in all selected manuscripts had been screened against the inclusion/exclusion criteria, as detailed above. This “backward snowballing” step helped to ensure that all the relevant literature was successfully found as part of this systematic review [23].

### 2.2. Bias Quality Assessment

The pre-clinical (animal) studies were assessed using SYRCLE’s risk of bias tool for animal studies [24], while a modified Newcastle–Ottawa Scale (NOS) for assessing the quality of nonrandomized studies in meta-analysis [25] was used to perform a quality assurance assessment of the selected clinical studies. The studies were assessed based on selection, comparability, and outcome, and were ranked from zero to four stars. Zero stars signified a lack of the information required and four stars signified that nothing else could be added and that the information perfectly matched the criteria.

The reviewers (G.D. and M.N.) divided the studies between them and any queries were resolved through discussions between them or when an agreement could not be reached, a third reviewer (B.G.) was consulted.

### 2.3. Data Extraction

For standardisation, a data extraction form was piloted in Excel using several selected studies with input from all reviewers. This included fields for study methodology, type of biological sample (i.e., blood), sample size, and outcomes, as well as the sensitivity and specificity of the biomarker. Both G.D. and M.N. performed the data extraction and any queries were resolved through discussions between them or when an agreement could not be reached, a third reviewer (B.G.) was consulted.

### 2.4. Meta-Analysis

RevMan was used to create box plots representing the pooled data for each biomarker. The likelihood ratios (LRs) were calculated for all the studies that presented their sensitivity and specificity. To enable this where specificity was presented as 100%, a small constant (0.01) was subtracted from all values for specificity. Values above 10 were considered strong evidence to rule in OT. Data analysis was performed using R Studio 2023.21.1 and the R package Metafor v4.6-0. A random effect meta-analysis for binary outcomes was performed, assigning data into one of two groups (OT vs. control) and two data outcomes (biomarker positive (+) and biomarker negative (−)). Data were grouped based on biomarker and the log odds ratio (OR), and sample variance was determined for each biomarker subgroup. A meta-regression model was used to calculate a combined log OR and test for subgroup differences.

## 3. Results

### 3.1. Screened Studies Selected for Systematic Review

This study was registered with the international prospective register of systematic reviews (PROSPERO) 2022 CRD42022370628 (Appendix A). The search returned 335 articles on blood biomarkers for OT (Figure 2). The articles were then screened by full text, with 23 meeting the eligibility criteria to be included in further analysis (Appendix A).

### 3.2. Quality Assurance

The 11 full-text versions of articles that focussed on pre-clinical (animal) studies were assessed using SYRCLE’s risk of bias tool for animal studies [24] (Table 2). None of the articles were excluded at this step and all met the requirements for data extraction. All had a good rating except Karatas Gurgun et al.’s 2017 study [26], which was deemed fair. The full-text versions of the 12 articles that focussed on clinical studies were assessed for eligibility and quality assurance using a modified Newcastle–Ottawa Scale (NOS) [25] (Table 3).

### 3.3. Meta-Analysis

#### 3.3.1. Biomarkers Studied Only in Animal Models

Articles that focussed on animal models used serum biomarker concentrations (Table 4) or absorbance units (AUs) [26,27,35]. The authors did not respond to requests to provide individual or pre-operative biomarker levels or the median and interquartile range of the results. Thus, the varied study designs and different units utilised for measuring each biomarker prevented a meta-analysis of the animal studies.

Plasma heat shock protein 70 (plasma HSP70; [29]) and pentraxin-3 (PTX3) were examined in animal models only. Plasma HSP70 belongs to a family of proteins that are released in response to cellular stress [49]. The authors randomised 21 Winstar albino rats into three groups, a torsion group, a sham operation group, and a group with no operation, and blood was sampled after 12 h. There was an increase in HSP70 in the torsion group (1.75 ng/mL) compared to both laparotomy (1.16 ng/mL) and control groups (1.19 ng/mL). Akman et al. [11] examined PTX3, which is a protein involved in the innate immune response [50]. The authors used 16 Sprague Dawley rats and induced ischaemia for 3 h. They found that the mean PTX3 was significantly higher in the torsion group (2.13 ng/mL) versus the control (1.07 ng/mL). CRP was also examined in one mouse model study [33], where the authors also found higher concentrations of CRP in the torsion group.

#### 3.3.2. Studies of Biomarkers in Clinical Trials

Of the articles that focussed solely on human studies, four articles explored the use of inflammatory ratios to diagnose OT [36,39,43,46]. Each assessed NLR, while Nissen et al. [43] also analysed the platelet-to-lymphocyte ratio (PLR). Ratios may give a more accurate indication of an acute inflammatory state [43] and while the cause of these ratios in patients experiencing OT has not been extensively studied, there is a possibility that they could aid with diagnosing and monitoring treatment response.

NLR was found to increase in OT compared to the controls [36,43,46], but there was a broad range of results (16–92% sensitivity [36,43]; 70.7–91% specificity [39,46]) **(**Figure 3), which may be explained by the study design. Aiob et al. [36] combined NLR with sonographic findings, while all other studies assessed NLR alone. In addition, Nissen et al. [43] focused on paediatric OT only (3 days old to 17 years and 8 months old), whereas all other studies included both adult and paediatric populations. Aiob et al. had a far larger sample size (n = 278; [36]) compared to Yilmaz et al. (n = 136; [46]), Ghimire et al. (n = 125; [39]), and Nissen et al. (n = 92; [43]). Only Yilmaz et al. [46] reported a cut-off value (2.44 NLR), which limited our capacity to perform a comparative analysis. Nissen et al. [43] also found that PLR had a lower specificity (82%) but a higher sensitivity (90%), the “strongest discriminatory accuracy”, and was independently predictive of OT. Recently, Delgado-Miguel et al. [51] described their retrospective multicentric case–control study of 75 patients with clinical and US suspicion of OT and 35 non-OT controls. They found a significant increase in inflammatory markers—leukocytes, neutrophils, the NLR ratio, and CRP. Their receiver operating characteristic (ROC) curve analysis showed that NLR had the highest area under the curve of 0.918 and maximum sensitivity (92.4%) and specificity (90.1%) at the cut-off of NLR = 2.57.

#### 3.3.3. Biomarkers Studied in Clinical and Pre-Clinical Studies

Certain biomarkers were studied in both pre-clinical animal models and in patients, including s-DD, IMA, IL-6, and signal peptide, CUB domain, and EGF-like domain-containing 1 (SCUBE1). s-DD is a well-established biomarker used to assess the presence of blood clot formation and fibrinolysis [52] and could be useful due to its association with OT [53]. The potential of s-DD was assessed in animal models [26,32] and humans [40,42,45]. Kart et al. [32] and Karatas Gurgun et al. [26] had similar study designs, and both found that DD increased in the torsion group. Karatas Gurgun et al. [26] found a far higher concentration of DD in rats 4 h after bilateral ovarian rotation compared to sham controls (*p* = 0.001), which may be accounted for by the fact that Kart et al. used a unilateral OT model. In addition, Karatas Gurgun et al. induced ischaemia for longer (4 h) than Kart et al., who used 2 h.

Gu et al. [40] and Incebiyik et al. [42] had a similar number of human participants (n = 28 and n = 34, respectively) compared to Topçu et al. [45] (n = 17). Topçu et al. [45] studied pregnant women with ovarian cysts who suffered from pelvic pain and found elevated s-DD levels in patients with surgically proven AT (77% versus 21%, *p* < 0.01). They noted that elevated s-DD and a cyst diameter >5 cm yielded the highest sensitivity (82%), while the presence of nausea and vomiting alongside elevated c-reactive protein (CRP) had the highest specificity (>85%). Plasma D-dimer (pls-DD) was significantly higher in women who were found to have an adnexal mass compared to those with benign ovarian cysts (*p* = 0.002). When a cut-off value of 0.65 µg/mL was used, the sensitivity and specificity for detecting AT was 71.4% and 85%, respectively [42] (Figure 3A). The likelihood ratio far exceeded the cut-off of 10 (Figure 3B), and the odds ratio for these three studies was also the highest of the biomarkers studied here (Figure 4). Gu et al. [40] described the use of sonographic markers (WS and pls-DD) and a laboratory index for AT in women presenting with a benign ovarian mass, abdominal pain, and clinically suspected AT. They observed an increase in WS and pls-DD levels in women with AT (cut-off level of 248 ng/mL) compared to those without (*p* < 0.01).

IMA is a biomarker used to assess ischemic conditions and measures alterations in the structure of albumin which occur in response to tissue ischaemia. Four studies assessed IMA only in animal models [26,27,35,54] and examined induced ischaemia for 3–4 h, although Karatas Gurgun et al. [26] and Lazăr et al. [54] took additional measurements over 24 h. Aside from Aran et al. [27], the papers took post-operative samples only. Results were overall promising, with three of the studies [27,35,54] finding a statistically significant increase in the IMA value. However, no statistically significant difference was identified by Karatas Gurgun et al. [26]. Guven et al. [41] showed that the oxidative stress marker, MDA, total oxidant status (TOS), and total antioxidant status (TAS) were all significantly higher in the torsion group (n = 14) compared to the control group (n = 20). The authors also found IMA had a relatively high sensitivity (90%), specificity (92.31%), and an LR = +11.7 (Figure 3B). Surprisingly, TAS, TOS, and the oxidative stress index (OSI) were highest in the control group that did not have OT.

IL-6 is a cytokine that contributes to the local inflammatory response in OT [55] and was the most observed biomarker in human [37,38,44,47] but not animal studies [31]. It has several roles in OT, such as the recruitment of immune cells [56] and angiogenesis, to restore blood flow in the torsed ovary [57]. A study by Reed et al. [44] found that IL-6 was elevated in patients with OT compared to non-surgical controls; however, there was no difference in IL-6 levels between patients with OT and appendicitis. In contrast, Cohen et al. [37] did find higher IL-6 levels in the torsion group. There was a substantial difference between the sensitivity reported by Zangene et al. (41.79%; n = 284 [47]) and Daponte et al. (92.3%; n = 37 [38]), although the specificity was closer (82.49% and 78.1%, respectively). This variation could be due to the larger sample size in Zangene et al.’s study, as well as the slight difference in cut-off values used in each study, 9.9 pg/mL in Zangene et al.’s work [47] versus 10.2 pg/mL in Daponte et al.’s work [38].

Uzun et al. in 2018 [34] found a statistically significant increase in mean SCUBE1 levels (*p* < 0.01) in rats that were subjected to bilateral OT and ischaemia lasting 24 h compared to rats that experienced bilateral OT and ovarian ischaemia for 8 h or the sham group that were only given a laparotomy procedure; however, Gunaydin et al. 2019 [30] found no difference between SCUBE1 levels after 4 h of unilateral torsion, suggesting this is not an early indicator of OT. Uyanikoglu et al. [58] noted a statistically significant increase in SCUBE1 (1.40 ng/mL versus 1.22 ng/mL) in patients with OT versus the control group, with a positive LR >10. Reported sensitivity was 80% and specificity was 93.3%.

**Figure 3 ijms-25-11664-f003:**
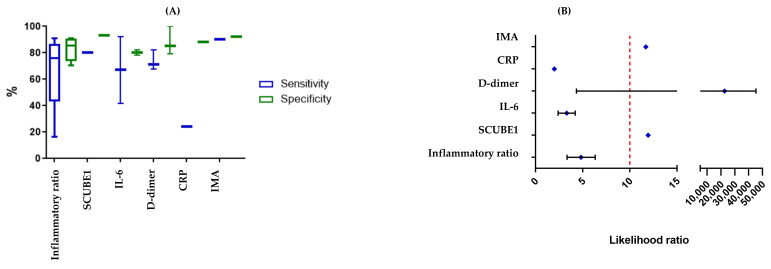
Sensitivity, specificity, and likelihood ratio of each biomarker as an indicator of OT or AT in studies of samples from human subjects. (**A**) Signal peptide, CUB domain, and EGF-like domain-containing 1 (SCUBE1) had the highest sensitivity, while IMA had the highest specificity. The interquartile (box), range (whisker), sensitivities (blue), and specificities (green) for pooled data on each biomarker are shown. (**B**) Likelihood ratios (LRs) with standard error of the mean calculated for each biomarker on studies that allowed it. The red dotted line indicates a positive LR of 10. LR above 10 are considered to provide strong evidence to rule in a diagnosis in most circumstances. Citations for markers were as follows inflammatory ratio [38,41,45,48], SCUBE1 [58], IL-6 [39,40,46,49], D-dimer [42,44,47], CRP [47] and IMA [43].

**Figure 4 ijms-25-11664-f004:**
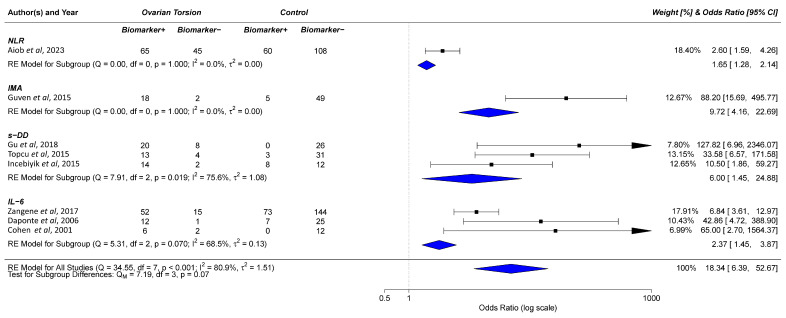
Analysis of biomarker detection in samples from human subjects with OT compared to the control group. The odds ratio is defined as the likelihood of an event in this study, in this case, elevated biomarker expression, occurring in the OT group compared to the control group. Q statistics for s-DD and IL-6 indicate a significant heterogeneity within these two subgroups, whilst I^2^ statistics of 75.6% and 68.5%, respectively, suggest the majority of heterogeneity observed in these subgroups is a result of true variance rather than sampling error. Individually, only s-DD had a statistically significant *p*-value (*p* = 0.019). Combined Q and I^2^ statistics confirm high heterogeneity and variance of true effect (Q = 34.55, I^2^ = 80.9%) with a statistically significant *p*-value (*p* ≤ 0.001). Both individual and combined ORs signify an increased likelihood of elevated biomarker expression in the OT group compared to the control group. Arrows indicate that the upper end of the 95% CI is outside the range of the scale bar. Citations used: Aiob et al., 2023 [36]; Guven et al., 2015 [41]; Gu et al., 2018 [40]; Topcu et al., 2015 [45]; Incebiyik et al., 2015 [42]; Zangene et al., 2017 [47]; Daponte et al., 2017 [38]; Cohen et al., 2001 [37].

## 4. Discussion

The reasons behind the delayed diagnosis of OT are complex; however, it is apparent that poor diagnostic tools are a key contributing factor to its impact on human health. The accuracy of a US diagnosis is highly dependent on operator skill and experience. It has been reported to be as low as 23–66% [59], which is exacerbated by the time-critical nature of an OT diagnosis and reliance on access to a skilled gynaecological sonography, which is often limited out-of-hours. There is no consensus as to whether immediate surgical intervention has consistently better outcomes than awaiting investigations [60]. This is due in part to the limitations of currently available diagnostic techniques, the avoidance of unnecessary surgical intervention, and uncertainty about the window prior to irreversible adnexal ischaemia. However, an accurate diagnostic test would aid clinicians in managing this rare but serious gynaecological emergency.

Eight biomarkers were identified in this review and overall, the sensitivity of the biomarkers across the eligible studies in humans ranged from 22% to 100%, and the specificity ranged from 60% to 100%. Promising biomarkers for the early prediction of OT included SCUBE1, s-DD, IL-6, IMA, and tumour necrosis factor-alpha (TNF-α). However, none thus far have been identified as clinically useful.

DD was widely used in clinical practise as a screening tool for venous thromboembolism (VTE) and to guide therapeutic anti-coagulation in unprovoked VTE [61]. Although the sensitivity reported in each paper was unremarkable, the specificity showed more promise. Notably, DD was the only biomarker that had a specificity of 100% [40], although this was not replicated in the two other articles assessing DD, which reported specificities of 78% and 84% (range 78–100%) [42,45]. The limitations of DD testing in current clinical practise are caused by variations due to age, pregnancy, VTE, malignancy, and the testing method, which have been widely discussed [61]. It should be noted that Gu et al. [40] excluded 26/94 participants due to this. Malignancy and pregnancy are both significant risk factors for OT and excluding these will elevate the specificity of DD but limit its clinical application. All of these factors would need to be addressed before DD integration into clinical practise; however, it is plausible that DD could be used as a tool to rule out OT.

SCUBE1, IL-6, and IMA also showed significant potential in human trials, although a larger sample size would be needed to confirm this. IL-6 had the highest sensitivity of any biomarker assessed in this review [38] and was raised in all 13 patients with proven OT, with values of ≥10.2 pg/mL, indicative of a 16-fold higher risk of having OT. SCUBE1 had a high specificity at 93.3%, second only to DD. Most importantly, SCUBE1 had a positive LR > 10. IMA was the only other biomarker that also had a positive LR > 10, making it a promising diagnostic biomarker. However, SCUBE1, IL-6, and IMA can be raised in other pathologies, especially chronic inflammatory and autoimmune diseases. Therefore, the use of these biomarkers alone may not be specific enough to identify OT.

Animal model results mirrored those seen in humans. Almost all the animal model papers found statistically significant increases in the biomarkers they were assessing. The exception to this was Gunaydin et al. [30], who assessed IMA. Plasma HSP70 and PTX3 were the only two biomarkers that were assessed in animal model papers alone [11,29]. Both studies found an increase in biomarker levels in the torsion group; however, it was notable that there was only one paper on each biomarker. There are, however, multiple papers on the role of PTX-3, IL-6, D-dimer, and CRP in other settings of urgent acute abdomen, such as acute appendicitis, one of the main diagnostic entities to consider in the differential diagnosis of ovarian torsion [62,63,64]. Ongoing research should continue to help identify more specific biomarkers. However, given the nature of the pathology and similarity of underlying mechanisms such as ischaemia and inflammation being present in many of the differential diagnoses, this may not be possible. At this stage, it seems unlikely that a non-invasive biomarker could be a gold standard diagnostic tool. Therefore, a combination of biomarkers or biomarkers and clinical findings may be a better way of identifying OT. This question was explored by Gu et al. [40], who found a sensitivity of 96.43% and a specificity of 100% when DD was combined with US. Further work should determine whether US can improve the sensitivity and specificity of other biomarkers that would enable an early OT diagnosis.

A notable finding from this literature search was the focus on venous blood biomarkers. This leaves urine and vaginal fluid as unexplored areas that could be used as a source of more sensitive and specific biomarker(s). It was also noteworthy that the studies amassed from this systematic literature review each had small sample sizes. This may reflect the infrequency by which OT occurs; however, larger studies will be needed for a fuller interpretation of the sensitivity and specificity of the biomarkers.

Limitations of this study include the heterogeneity of the objectives, inclusion criteria, participant recruitment, and methodology of the human studies selected by this systematic review. In addition, the methodology and study design varied notably amongst the animal model studies. The absence of pre-operative measurements in six studies [26,30,31,34,35,54] made it difficult to assess the extent of the increase in each biomarker. The lack of standardisation between the methods was also a limitation of this study, with studies varying in the degree of adnexal twisting, whether it was unilateral or bilateral, and the length of time that ischaemia was induced for.

## 5. Conclusions

The standardisation of the study methodology would enable a more reliable identification of potential biomarkers for the early detection of OT. This review sought to identify the most promising biomarkers for the early detection of OT, and this is the first systematic review to evaluate the state of the art in non-invasive biomarkers for the diagnosis of OT as examined in animal models and patients. IMA, DD, and SCUBE1 show promise as markers with a pooled LR > 10 in humans with OT. However, further well-designed trials are needed to identify and evaluate individual markers or diagnostic panels to help clinicians manage this important organ-threatening condition.

## Figures and Tables

**Figure 1 ijms-25-11664-f001:**
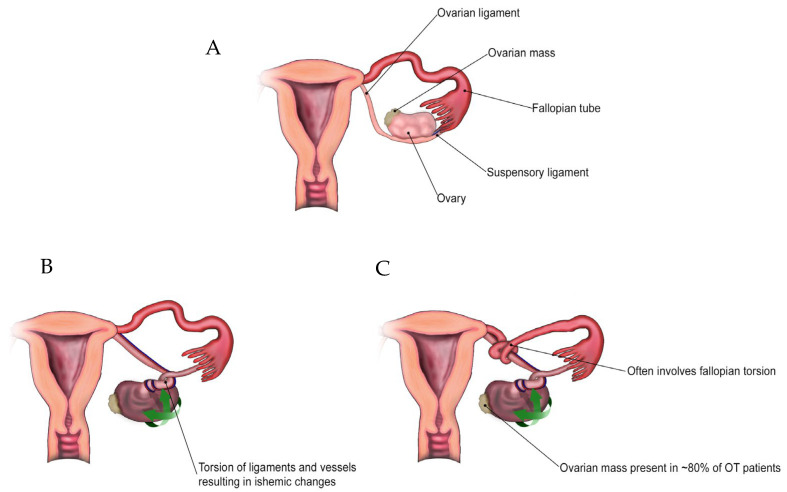
Mechanisms of ovarian torsion (OT). (**A**) Pre-torsion anatomy showing ovarian mass; (**B**) OT with torsion of ovarian/suspensory ligaments and ovarian vessels; (**C**) Adnexal torsion with additional torsion of the fallopian tube. Green arrows indicate direction of movement in two of the three possible dimensions.

**Figure 2 ijms-25-11664-f002:**
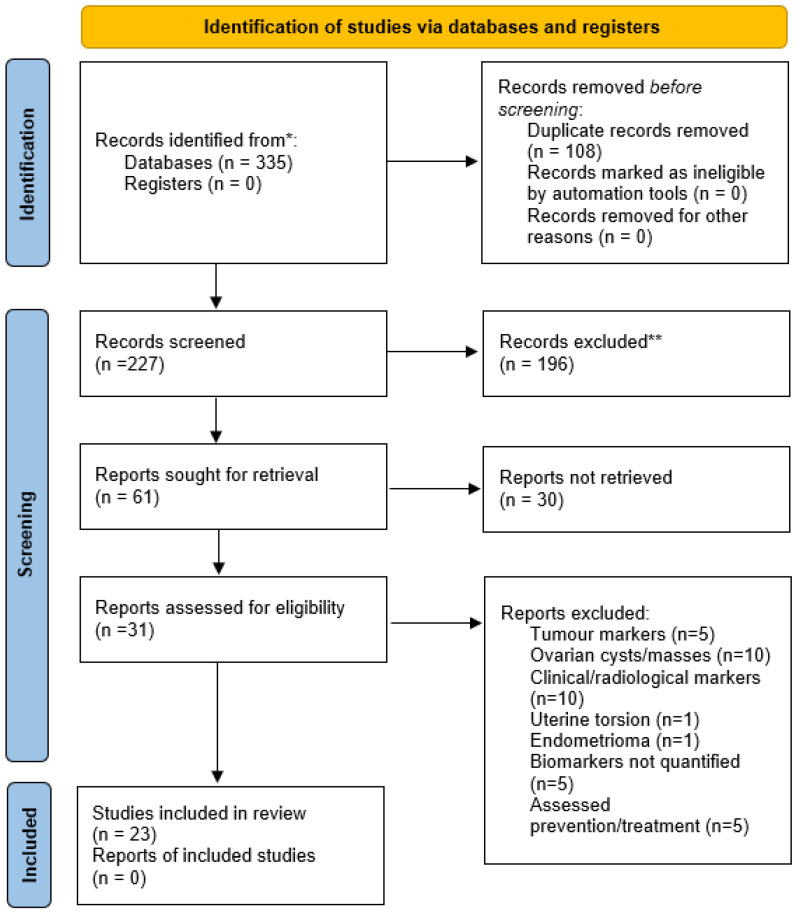
Preferred Reporting Items for Systematic reviews and Meta-Analyses (PRISMA) flow diagram. Searches of the databases PubMed, Medline, Scopus, Cochrane, and CINAHL identified 335 articles that appeared to relate to the project title. The removal of any duplicated articles was followed by the selection of articles based on their title and/or abstract. Articles were then screened against the inclusion and exclusion criteria, and papers were excluded due to the lack of quantified biomarkers for OT or assessing biomarkers for conditions that were not OT. After the final full text screening, the 24 articles that remained were deemed to meet the inclusion criteria and fit the eligibility screening. * Records identified in five online databases (PubMed, Medline, Scopus, Cochrane, and CINAHL) from inception until 1 October 2023; ** Records excluded based on the prespecified inclusion/exclusion criteria.

**Table 1 ijms-25-11664-t001:** Exclusion/inclusion criteria.

Inclusion	Exclusion
Primary literature sources with reviews kept in until screening is completed to facilitate the benefits of reverse snowballing	Biomarkers are not directly quantified (qualitative biomarkers)
Human/clinical studies and animal models	Cell lines
Patients diagnosed with OT	Disease other than OT
Quantitative biomarkers	Biomarkers for ovarian cancer
Have a quantified non-invasive blood biomarker expression which predicts OT	
All ages	
All geographical locations	
All publication dates	

**Table 2 ijms-25-11664-t002:** SYRCLE’s risk of bias tool for the assessment of pre-clinical animal studies prior to further analysis.

Citation	D1	D2	D3	D4	D5	D6	D7	D8	D9	D10
Akman et al. 2016 [11]	+	+	+	?	+	?	+	+	?	+
Aran et al. 2010 [27]	+	+	+	?	+	+	+	+	?	+
Bakacak et al. 2015 [28]	+	+	+	?	+	+	+	+	?	+
Cilgin et al. 2019 [29]	+	+	?	?	?	?	?	+	+	+
Gunaydin et al. 2019 [30]	+	+	?	?	?	?	?	+	+	+
Karakoc-Sokmenseur et al. 2016 [31]	+	+	+	?	+	+	+	+	?	+
Karatas Gurgun et al. 2016 [26]	+	+	?	?	?	?	?	+	+	+
Kart et al. 2011 [32]	+	+	+	?	+	+	+	+	?	+
Lazăr et al. 2019 [33]	+	+	?	?	?	?	?	+	+	+
Uzun et al. 2018 [34]	+	+	?	?	?	?	+	+	+	+
Yildrim et al. 2016 [35]	+	+	?	?	?	?	+	+	+	+

D1 assesses sequence generation, D2 assesses baseline characteristics, D3 assesses allocation concealment, D4 assesses random housing, D5 assesses blinding (caregivers), D6 assesses random outcome assessment, D7 assesses blinding (outcome assessors), D8 assesses incomplete outcome date, D9 assesses selective outcome reporting, and D10 assesses any other sources of bias. The + indicates that this assessment criteria was completed by the authors and ? shows no mention of this source of bias in the paper.

**Table 3 ijms-25-11664-t003:** NOS criteria of selection, comparability, and outcome were applied to the selected clinical studies. Studies that had zero stars would have been excluded, as these would have been deemed to not contain the depth of information required to be included in this review.

Paper	Selection	Comparability	Outcome
Aiob et al. 2023 [36]		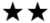	
Cohen et al. 2001 [37]		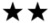	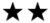
Daponte et al. 2006 [38]		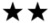	
Ghimire et al. 2023 [39]		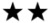	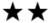
Gu et al. 2018 [40]		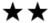	
Guven et al. 2015 [41]		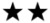	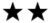
Incebiyik et al. 2015 [42]		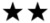	
Nissen et al. 2019 [43]		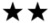	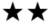
Reed et al. 2011 [44]		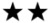	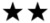
Topçu et al. 2015 [45]		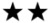	
Yilmaz et al. 2015 [46]		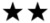	
Zangene et al. 2017 [47]		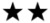	

The thresholds for converting the NOS to the Agency for Healthcare Research and Quality (AHRQ) standards [48] of good, fair, or poor were applied as follows: good quality—three to four stars in the selection domain AND one to two stars in the comparability domain AND two to three stars in the outcome domain; fair quality—two stars in the selection domain AND one to two stars in the comparability domain AND two to three stars in the outcome domain; poor quality—zero to one star in the selection domain OR zero stars in the comparability domain OR 0/1 star in the outcome domain.

**Table 4 ijms-25-11664-t004:** Animal studies of OT and AT.

Article	Primary Data				*p*-Value	Secondary Data
Aran et al. 2010 [27]	Pre-operative IMA (AU)				Similar in both groups	Group 2 had increased follicular degeneration
	Post-operative IMA (AU)				0.05	
	Sham operation: 0.191 ± 0.034		Torsion model: 0.277 ± 0.089			
Akman et al. 2016 [11]	Pentraxin-3 (PTX3)		Sham operation (ng/mL)	Torsion model (ng/mL)		
	Pre-operative levels (ng/mL)		1.05 ± 0.20	1.09 ± 0.28	>0.05	Higher follicular degeneration in group
	Post-operative levels (ng/mL)		1.07 ± 0.22	2.13 ± 0.49	0.001	
Bakacak et al. 2015 [28]			Sham operation	Torsion model		Higher follicular cell degeneration in group 2
	CRP levels	Pre-operative	0.36 ± 0.04	0.36 ± 0.04	0.214	
	(mg/L)	Post-operative	0.39 ± 0.06	0.91 ± 0.18	<0.001	
Çilgin et al. 2017 [29]	Heat shock protein 70 (hsp-70; ng/mL)				No statistical difference between pre-operative levels (*p* = 0.966); *p* = 0.001 for post-operative levels	
		Torsion model (Group 1)	Sham operation (Group 2)	No operation (Group 3)		
	Pre-operative	1.19 (±0.13)	1.18 (±0.78)	1.15 (±0.49)		/
	Post-operative	1.75 (±0.25)	1.16 (±0.99)	1.19 (±0.11)	Statistically significant difference between Group 1 and 2 (*p* = 0.002), 1 and 3 (*p* = 0.002), but not 2 and 3 (*p* = 0.561)	
Gunaydin et al. 2019 [30]			Control	Torsion		Increased vascular congestion and haemorrhage in the torsion group
	SCUBE1 (ng/mL)		1.83 ± 0.16	1.82 ± 0.18	0.987	
	Superoxide dismutase (SOD) (U/mL)		5.33 ± 0.44	5.98 ± 0.45	0.33	
	Malondialdehyde (MDA) (mmol/L)		25.81 ± 2.16	33.83 ± 2.78	0.039	
	Total antioxidant status (TAS; mmmol Trolox Evuiv/L)		0.92 ± 0.01	1.04 ± 0.08	0.244	
Karakoc-Sokmensuer et al. 2016 [31]	No change to mean plasma IL-6 (pg/mL)				0.584	Total tissue damage was similar across groups
Karatas Gurgun et al. 2017 [26]		Sham operation	4 hr torsion	24 hr torsion		Increased follicular cell degeneration in the torsion groups
	IMA (ng/mL)	0.59 ± 0.06	0.58 ± 0.1	0.71 ± 0.14	0.064	
	DD (ng/mL)	250.71 ± 71.95	1740.20 ± 913.94	474.36 ± 222.4	0.001	
Kart et al. 2011 [32]	DD (mg/L)	Sham operation		Torsion model		Greater follicular cell degeneration in group 2
	Pre-operative plasma levels	0.5963 ± 0.2047		0.6344 ± 0.1348	0.815	
	2 h after OT	1.2267 ± 0.3099		0.6213 ± 0.2346	0.001	
	Mean difference	0.0250 ± 0.2660		0.5922 ± 0.3001	0.001	
Lazăr et al. 2019 [33]				IMA levels (μmol/L)		
	Control			402.370 ± 2.732	0.003	
	Sham operation			418.472 ± 1.854		
	3 h torsion			478.359 ± 5.218	<0.001	
	3 h torsion, 1 h simple reperfusion			490.024 ± 3.376		
	3 h ischaemia, 1 h controlled reperfusion			452.564 ± 3.096		
	3 h ischaemia, 24 h simple reperfusion			483.370 ± 1.550		
	3 h ischaemia, 24 h controlled reperfusion			454.207 ± 0.878		
Uzun et al. 2018 [34]	Post-operative	Group 1: sham operation	Group 2: torsion model + ischaemia >8 h	Group 3: torsion model + ischaemia >24 h	Group 1: 0.004 Group 2: 0.150 Group 3: 0.016	Increased follicular degeneration in the ischemic groups
	SCUBE1 (ng/mL)	51.12 ± 17.04	71.83 ± 20.53	132.85 ± 51.18		
Yildirim et al. 2016 [35]	Median post-operative IMA levels (ABSU)		Torsion group	Control group	0.001 for the difference between groups	No pathological change in the control group, but pathological change present in the OT group
			921 (range: 870.0–966.00)	853 (range: 783–869)		

## Data Availability

The original data presented in the study are openly available on zenodo.org at https://zenodo.org/records/10072093 in the Appendix A.

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
