# Peer review of "Are There Non-Invasive Biomarker(s) That Would Facilitate the Detection of Ovarian Torsion? A Systematic Review and Meta-Analysis"

_ijms, 2024, doi:10.3390/ijms252111664_

Round 1
Reviewer 1 Report
Comments and Suggestions for Authors
Although this review addresses an interesting topic, the methodology applied is incorrect on multiple levels, and the review does not meet the expected quality standards. My comments are attached.
1. The English needs revision by a native speaker. There are incorrect expressions and inappropriate terms (e.g., line 140, type of specimen > biological sample; the search amassed 335 articles > the search returned 335 articles).
2. A major limitation is Mixing animal and human studies in a review. In fact, there are specific bias assessment scales for animal studies.
3. The abstract should be summarised. Inclusion and exclusion criteria are unnecessary and should be detailed in the manuscript.
4. There are confusing phrases in the abstract
a. "IL-6 had the highest sensitivity of any biomarker identified in this review and was raised in all 13 patients with proven OT, with values of ≥10.2pg/ml, indicative of a 16 times higher risk of having OT." Were 13 patients included in this analysis, or is this a typo? This value is insufficient. Does this Odds ratio correspond to a meta-analytic model? Not specified.
5. The abstract should be structured (introduction, material and methods, results, and conclusions).
6. The graphical abstract is not adequately understood. On the one hand, it provides a generic anatomical image, which is not necessary, and on the other hand, it provides a graph of the confluence of biochemical markers between murine and human species without explanation. This does not correspond to the submitted manuscript.
7. The introduction is incorrectly formulated and insufficient. It should not be an illustrative text on ovarian torsion but an introduction aimed at the potential role of biomarkers in its early diagnosis.
8. The authors do not discuss how the presented markers have demonstrated utility in similar diagnostic settings. There are multiple papers on the role of PTX-3, IL-6, D-dimer, and CRP in other settings of urgent acute abdomen, such as acute appendicitis, one of the main diagnostic entities to consider in the differential of ovarian torsion. I recommend a detailed review of the literature. I have attached some illustrative references
Di Mitri M, Parente G, Bonfiglioli G, Thomas E, Bisanti C, Cordola C, Vastano M, Cravano S, Collautti E, Di Carmine A, D'Antonio S, Gargano T, Libri M, Lima M. IL-6 Serum Levels Can Enhance the Diagnostic Power of Standard Blood Tests for Acute Appendicitis. Children (Basel). 2022 Sep 20;9(10):1425. doi: 10.3390/children9101425. PMID: 36291361; PMCID: PMC9600576.
Arredondo Montero J, Antona G, Bronte Anaut M, Bardají Pascual C, Ros Briones R, Fernández-Celis A, Rivero Marcotegui A, López-Andrés N, Martín-Calvo N. Diagnostic performance of serum pentraxin-3 in pediatric acute appendicitis: a prospective diagnostic validation study. Pediatr Surg Int. 2022 Dec 1;39(1):27. doi: 10.1007/s00383-022-05289-7. Erratum in: Pediatr Surg Int. 2022 Dec 13;39(1):49. doi: 10.1007/s00383-022-05342-5. PMID: 36454367; PMCID: PMC9713741.
Tayebi A, Olamaeian F, Mostafavi K, Khosravi K, Tizmaghz A, Bahardoust M, Zakaryaei A, Mehr DE. Assessment of Alvarado criteria, ultrasound, CRP, and their combination in patients with suspected acute appendicitis: a single centre study. BMC Gastroenterol. 2024 Jul 31;24(1):243. doi: 10.1186/s12876-024-03333-5. PMID: 39085761; PMCID: PMC11289915.
9. The authors should also discuss the intrinsic characteristics of these markers as acute phase reactants.
10. Authors should briefly mention the particularities of pediatric populations and the existence of recent specific studies in this field.
Delgado-Miguel C, Arredondo-Montero J, Moreno-Alfonso JC, San Basilio M, Peña Pérez R, Carrera N, Aguado P, Fuentes E, Díez R, Hernández-Oliveros F. The Role of Neutrophyl-to-Lymphocyte Ratio as a Predictor of Ovarian Torsion in Children: Results of a Multicentric Study. Life (Basel). 2024 Jul 18;14(7):889. doi: 10.3390/life14070889. PMID: 39063642; PMCID: PMC11277755.
11. The Flow diagram used is out of date. Use the latest version of PRISMA.
12. The NOS scale is a scale for studies conducted in humans. What version have you used for adaptation to animal studies? Is this version validated?
13. The meta-analytic model in Figure 2 is not logical. A DTA meta-analysis is appropriate for a diagnostic performance study. If another model is used, such as an inverse variance, it should be justified. What do the events per group represent in the case of the meta-analyses in Figure 2? The authors proposed nonsense, such as two meta-analytic models WITH ONE STUDY (D, E). Please consult an experienced statistician.
14. Figure 3 does not provide sufficient information about the data represented. The studies' sensitivities and specificities should be represented by a specific DTA forest plot or an sROC curve using a hierarchical logistic regression model. The authors are evaluating the diagnostic performance of biomarkers in pathology. The Cochrane guidelines define this as a diagnostic test accuracy review, and the authors should follow the relevant steps in this regard.
15. Table 1 needs to be revised. The measures of central tendency and dispersion employed have not been clarified. In some cases, the symbol ± is used; in others, the symbol + is used. In none of the studies is the AUC curve provided, which defines diagnostic performance (and not the p-value for comparison between groups). The authors also mix up concepts by including preoperative and postoperative values. According to the title and the prospective registry, this review is limited to ovarian torsion detection (diagnosis). Data on the sociodemographics of the patients, the nature of the studies (prospective/retrospective), etc., are not included.
Comments on the Quality of English LanguageExtensive editing of English language required.
Reviewer 2 Report
Comments and Suggestions for Authors
The authors present a systematic review and meta-analysis of different biomarkers to differentiate ovarian torsion.
It is a very interesting subject, since the diagnosis of this pathology is not always simple, and the question of whether or not to intervene in these patients is something we encounter on a daily basis.
The methodology is adequate, with well-structured graphs and tables relevant to this type of study.
I strongly recommend including a multicenter study with more than 100 patients, which was published a few months ago, which analyzes precisely this.
Delgado-Miguel C et al. The Role of Neutrophyl-to-Lymphocyte Ratio as a Predictor of Ovarian Torsion in Children: Results of a Multicentric Study. Life (Basel). 2024 Jul 18;14(7):889. doi: 10.3390/life14070889
Reviewer 3 Report
Comments and Suggestions for Authors
In the present work, Doherty et al. try to review the non-invasive biomarker(s) that would facilitate the detection of ovarian torsion. In this manuscript, two independent reviewers performed systematic searches of five literature databases to determine non-invasive detection of OT. Some proteins may be used as promising biomarkers, and IL-6 had the highest sensitivity among these biomarkers in this review. However, there are some questions that should be explained.
Major concerns
1. Graphical Abstract for ovarian torsion is not right. There is no substantial connection between the ovaries and the fimbriae of the oviduct. Please see ‘https://my.clevelandclinic.org/health/diseases/ovarian-torsion’. Therefore, it should be revised.
2. Causes for ovarian torsion should be added. In addition, a fine figure for the mechanism of ovarian torsion development may be needed.
3. English grammar and writing style should be checked and revised throughout the manuscript. I suggest that it is supported by a professional English language proofreading service.
Minor concerns
1. The writing style of Abstract section is not suitable for IJMS, which should be rewritten.
2. Keywords, delete ‘systematic review’.
3. Graphical Abstract, figure legend should be added.
4. There is no 3.3a. In addition, 3.3b, 3.3c, 3.3d (two 3.3d) should be renumbered.
5. Line 189, ‘tool [46](Figure 2).’. a blank space should be added.
6. Figure 2, P = 0.02, or p = 0.02. p should be in italic. Check it throughout this manuscript.
7. Line 248, change ‘1.19ng/ml’ to ‘1.19 ng/ml’. Check it throughout this manuscript.
8. Line 262, ‘[11,12])(Figure 3A)’. a blank space should be added. Check it throughout this manuscript.
9. Line 327, ‘Uzun et al. 2018’ or ‘Uzun et al., 2018’. Check it throughout this manuscript.
10. Lines 403-405, ‘Another problem was the lack of standardisation between the methods used ie there was variation between the degree of adnexal twisting, whether it was unilateral or bilateral, and the length of time that ischaemia was induced for.’. These sentences should be revised.
Comments on the Quality of English LanguageExtensive editing of English language required.
Reviewer 4 Report
Comments and Suggestions for Authors
In this study the authors compared many non-invasive biomarkers for the detection of OT, both in humans and in animals, and showed that, in humans, the most promising biomarkers for the early prediction of OT included s-DD, interleukin-6 (IL-6), IMA and tumour necrosis factor-alpha (TNF-alpha), whereas for animals, they were ischemia modified albumin (IMA), serum D-dimer (s-DD), heat shock protein-70 (hsp-70), Pentraxin-3 (PTX-3) and c-reactive protein (CRP), being IMA and s-DD common between the two groups. However, though the sensitivity and the specificity were high in many of them (except for CRP), the authors underlined that none of them would be used as useful tools for the early prediction of the OT, being common parameters also for other diseases, in absence of further confirmations, such as the preliminary US.
The authors have performed a scrupulous screening of the existing literature, highlighting the difficulty in finding cases of OT described with the necessary complete anamnesis reporting the analyses performed (blood tests, parameters, etc.), as well as the absence of further pathologies worsening the clinical picture, and therefore excluded from the study. Furthermore, they performed a well-described statistical analysis consistent with the purpose of the review, clear and logically developed
Round 2
Reviewer 3 Report
Comments and Suggestions for Authors
Thanks for author’s responses. However, some questions still should be explained.
1. Previous Graphical Abstract for ovarian torsion should be revised. However, it has been deleted in the revision.
2. A fine figure for the mechanism of ovarian torsion development should be added. The existence of very good ones in the literature already is not a right cause.
3. English grammar and writing style should still be checked and revised throughout the manuscript. For example, line 168, ‘[24](Table 2)’.
Comments on the Quality of English LanguageModerate editing of English language required.
Author Response
Dear Reviewer,
Thank you for your thoughtful comments. These have been dealt with as follows:-
- Previous Graphical Abstract for ovarian torsion should be revised. However, it has been deleted in the revision.
The graphical abstract has been removed from the text, but revised as suggested.
2. A fine figure for the mechanism of ovarian torsion development should be added. The existence of very good ones in the literature already is not a right cause.
Figure 1 has been added as suggested.
3. English grammar and writing style should still be checked and revised throughout the manuscript. For example, line 168, ‘[24](Table 2)’.
This has been checked and revised.